# Estimating ectopic beat probability with simplified statistical models that account for experimental uncertainty

**Qingchu Jin, Joseph L. Greenstein[ID], Raimond L. Winslow[ID]***

Department of Biomedical Engineering and Institute for Computational Medicine, Johns Hopkins University, Baltimore, Maryland, United States of America

* rwinslow@jhu.edu

**Data Availability Statement:** The model and code is available on Github: https://github.com/JHU-Winslow-Lab/Ectopic-beat-paper-code.git.

**Funding:** Q.J, J.L.G and R.L.W. are funded by National Institutes of Health R01HL105239.

## Abstract

Ectopic beats (EBs) are cellular arrhythmias that can trigger lethal arrhythmias. Simulations using biophysically-detailed cardiac myocyte models can reveal how model parameters influence the probability of these cellular arrhythmias, however such analyses can pose a huge computational burden. Here, we develop a simplified approach in which logistic regression models (LRMs) are used to define a mapping between the parameters of complex cell models and the probability of EBs (P(EB)). As an example, in this study, we build an LRM for P(EB) as a function of the initial value of diastolic cytosolic $Ca^{2+}$ concentration ($[Ca^{2+}]_i^{ini}$), the initial state of sarcoplasmic reticulum (SR) $Ca^{2+}$ load ($[Ca^{2+}]_{SR}^{ini}$), and kinetic parameters of the inward rectifier $K^+$ current ($I_{K1}$) and ryanodine receptor (RyR). This approach, which we refer to as arrhythmia sensitivity analysis, allows for evaluation of the relationship between these arrhythmic event probabilities and their associated parameters. This LRM is also used to demonstrate how uncertainties in experimentally measured values determine the uncertainty in P(EB). In a study of the role of $[Ca^{2+}]_{SR}^{ini}$ uncertainty, we show a special property of the uncertainty in P(EB), where with increasing $[Ca^{2+}]_{SR}^{ini}$ uncertainty, P(EB) uncertainty first increases and then decreases. Lastly, we demonstrate that $I_{K1}$ suppression, at the level that occurs in heart failure myocytes, increases P(EB).

## Author summary

An ectopic beat is an abnormal cellular electrical event which can trigger dangerous arrhythmias in the heart. Complex biophysical models of the cardiac myocyte can be used to reveal how cell properties affect the probability of ectopic beats. However, such analyses can pose a huge computational burden. We develop a simplified approach that enables a highly complex biophysical model to be reduced to a rather simple statistical model from which the functional relationship between myocyte model parameters and the probability of an ectopic beat is determined. We refer to this approach as arrhythmia sensitivity analysis. Given the efficiency of our approach, we also use it to demonstrate how uncertainties in experimentally measured myocyte model parameters determine the uncertainty in ectopic beat probability. We find that, with increasing model parameter uncertainty, the

Website: https://grants.nih.gov/grants/funding/r01.htm. Q.J. is supported by David C. Gakenheimer Fellowship. Website: https://icm.jhu.edu/2013/02/11/test-238/#.YEvpbZ1KhPY. The funders had no role in study design, data collection and analysis, decision to publish, or preparation of the manuscript.

**Competing interests:** The authors have declared that no competing interests exist.

uncertainty in probability of ectopic beat first increases and then decreases. In general, our approach can efficiently analyze the relationship between cardiac myocyte parameters and the probability of ectopic beats and can be used to study how uncertainty of these cardiac myocyte parameters influences the ectopic beat probability.

This is a *PLOS Computational Biology* Methods paper.

## Introduction

Delayed after-depolarizations (DADs) are spontaneous depolarizations that occur during diastole [1]. When their amplitude is sufficiently large, DADs can trigger action potentials (APs) known as ectopic beats (EBs) [2]. Under certain conditions, for example in the setting of catecholaminergic polymorphic ventricular tachycardia, DADs can also trigger large scale cardiac arrhythmias [3]. Computational modeling of EBs and DADs has provided insights into the mechanisms by which they can trigger arrhythmias in the heart [4,5].

We seek to understand how variations in the underlying biophysical properties or physiological state of the myocyte influence the probability of occurrence of EBs. To do this, we use a three-dimensional spatial model of the ventricular myocyte developed by Walker et al [5] in which the fundamental events governing intracellular calcium ($Ca^{2+}$) dynamics are modeled stochastically. This model has previously been shown to reproduce realistic $Ca^{2+}$ waves, DADs, and ectopic beats driven by stochastic gating of L-type $Ca^{2+}$ channels (LCCs) and sarcoplasmic reticulum (SR) $Ca^{2+}$ release channels (ryanodine receptors, RyRs).

The complexity of this stochastic model makes it challenging to perform the many repeated simulations needed to estimate the probability of ectopic beats, denoted P(EB), as a function of underlying model parameters. Here we explore an approach designed for estimating P(EB) that is computationally efficient. To do this we leverage a technique first developed by Sobie et al [6,7] and apply it to formulate a logistic regression model (LRM) that maps myocyte model inputs (MMIs), which refers collectively to myocyte model parameters as well as state variable initial conditions, to probabilities of cellular arrhythmias. This approach, which we refer to as arrhythmia sensitivity analysis, allows for evaluation of the relationship between these event probabilities and their associated MMIs.

Use of an LRM to map MMIs to the probability of cellular arrhythmias also enables analyses of how experimental uncertainty in the measurement of MMIs influences estimates of these probabilities. One mainstream method for experimental uncertainty analysis is Monte Carlo simulation [8]. It has been used extensively in system biology and myocyte modeling [9–11]. However, such methods require many repeated model simulations, an approach which can become computationally demanding when using complex models of the myocyte such as that of Walker et al. [5]. LRMs also provide a computationally efficient approach for performing uncertainty analyses.

In this study, we develop a LRM model, mapping 4 MMIs (initial state of diastolic cytosolic $Ca^{2+}$ concentration ($[Ca^{2+}]_i^{ini}$), initial state of SR $Ca^{2+}$ load ($[Ca^{2+}]_{SR}^{ini}$), conductance of the inward rectifier current $I_{K1}$ ($G_{K1}$), and RyR opening rate ($k_{RyR}^+$)) to P(EB). We also investigate the role of uncertainty in $[Ca^{2+}]_{SR}^{ini}$ on P(EB). It surprisingly reveals that as $[Ca^{2+}]_{SR}^{ini}$ uncertainty increases, P(EB) uncertainty first increases (from a sharp unimodal distribution to a broad uniform-like distribution) and then decreases (toward a bimodal distribution with P

(EB) concentrated at 0 or 1). Investigation of $G_{K1}$ uncertainty showed that P(EB), with $G_{K1}$ at reduced levels associated with heart failure, is significantly higher than that at normal $G_{K1}$ density.

## Results

### Modeling ectopic beat probability

An LRM was formulated to capture the quantitative relationship between P(EB) and 4 MMIs ($[Ca^{2+}]_i^{ini}$, $[Ca^{2+}]_{SR}^{ini}$, $G_{K1}$ scaling factor ($G_{K1}\_sf$) and $k_{RyR}^+$ scaling factor ($k_{RyR}^+\_sf$)), where $[Ca^{2+}]_i^{ini}$ and $[Ca^{2+}]_{SR}^{ini}$ are the initial values of cytosolic and SR $Ca^{2+}$ concentration, respectively, for each model simulation (i.e. realization). For each realization, the myocyte model simulation was run incorporating these 4 MMI values for up to 800 ms. Besides $[Ca^{2+}]_i^{ini}$, $[Ca^{2+}]_{SR}^{ini}$, all other initial states of the myocyte model were set to values consistent with the diastolic state. To study ectopic beats and DADs, experiments generally have to create conditions that will favor the generation of DADs. We therefore performed simulations under conditions that favor the generation of DADs and ectopic beats. We used a modified version of the Walker et al. model [5]. A novel $Na^+$—$Ca^{2+}$ exchanger (NCX) model developed by Chu et al. [12] was incorporated into this model. Adjustments were made to the $Ca^{2+}$ flux rate from submembrane to cytosol, the conductance of the rapid delayed rectifier $K^+$ current and the density of the $Ca^{2+}$ dependent $Cl^-$ current to ensure a realistic action potential duration (~290ms) and $Ca^{2+}$ transient amplitude (~800nM) [13]. A detailed description of model modifications is provided in Methods. Myocyte model simulations were performed under the β-adrenergic stimulation condition described in Walker et al [5], with the exception that $k_{RyR}^+$ was increased 4-fold compared to its control value based on experimental data for heart failure [14] and to increase the propensity for EBs. A detailed description of simulation protocol is described in Methods and S1 Text. Because of the stochastic nature of the myocyte model, each realization may or may not generate a DAD or an ectopic beat. We defined a biophysically meaningful region of interest over which each of these parameters are varied and within which the LRM is built, where the range for $[Ca^{2+}]_i^{ini}$ is 100nM– 300nM, the range for $[Ca^{2+}]_{SR}^{ini}$ is 300 μM–700 μM, the range for $G_{K1}\_sf$ is 0–1 and the range for $k_{RyR}^+\_sf$ is 0.5–1.5. Fig 1A shows five examples each of myocyte model simulated ectopic beats (red), DADs (blue), and no spontaneous depolarization events (green). Since ectopic beats consistently exhibit membrane potential waveforms that exceed 0 mV whereas DADs do not, a threshold of 0 mV was used to detect such events in each realization. Fig 1B demonstrates the relationship between this region of interest and P(EB) for both the underlying myocyte model and the LRM, where $\underline{B}^T\underline{P}$ (the argument of the logistic regression function) is the weighted ($\underline{B}$) summation of LRM features ($\underline{P}$). The nonlinear logistic relationship defines three domains of interest: transition domain, lower plateau domain, and upper plateau domain. The transition domain is a region in which P(EB) is a sensitive function of $\underline{B}^T\underline{P}$, corresponding to the steep part of the curve in Fig 1B. The lower plateau domain and upper plateau domain are regions where P(EB) is relatively insensitive to MMI perturbation, with P(EB) = ~0 in the lower plateau domain and P(EB) = ~1 in the upper plateau domain. To build the LRM, we sampled 200 MMI sets from the region of interest and ran 100 realizations for each set from which we obtain P(EB). Since the transition domain is relatively narrow and P(EB) is a very steep function within it, we have developed a two-stage sampling strategy to ensure adequate sampling of this domain (see Methods and S1 Text for details). The 200 MMI sets and their corresponding P(EB)s were used to train the LRM. Once determined, the LRM enables direct calculation of P(EB) for any MMI set without the need to simulate the underlying complex stochastic three-dimensional myocyte model. Fig 1B shows the fidelity with which the LRM model (blue line) reproduces P(EB) estimated using

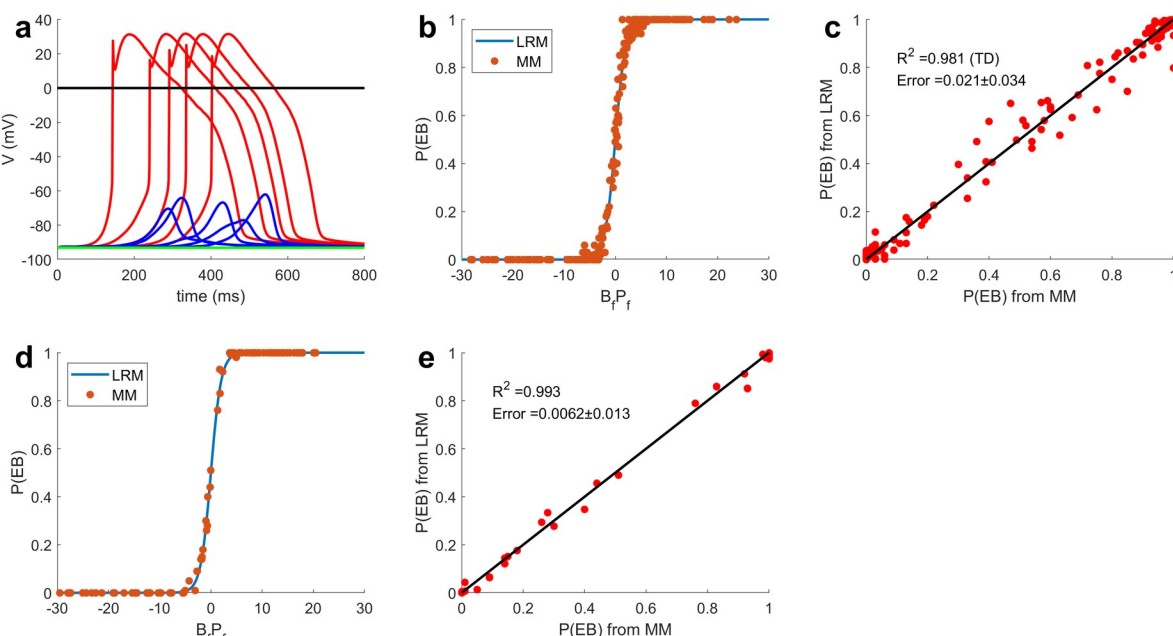

**Fig 1. Ectopic beat study.** (A) Simulations of ectopic beats (red),simulations that generate DADs without induced ectopic beats (blue) and simulations that exhibit no spontaneous depolarization events (green). (B) Comparison between LRM-predicted P(EB)s (blue) and myocyte model-generated actual P(EB)s (red) as a function of $\underline{B}^T\underline{P}$, the argument of the logistic equation (S3 Eq for the training set). (C) Direct comparison between LRM-predicted P(EB)s (y-axis) and myocyte model-generated actual P(EB)s (x-axis) for the training set. (D) and (E) show the same analyses described in (B) and (C), respectively, for the independent test set. $R^2$ is calculated only from MMI sets within the transition domain, LRM: logistic regression model, MM: myocyte model.

the myocyte model (red points) for the training data set (200 MMI sets) as a function of $\underline{B}^T\underline{P}$. Fig 1C shows the predicted P(EB) from the LRM versus P(EB) computed from the myocyte model simulations. The LRM performs well in predicting P(EB) ($R^2$ = 0.981 for the 102 MMI sets inside the transition domain, mean prediction error of 0.021 ± 0.034 (see Methods and Eq 2)), despite the highly complex nonlinear properties of the myocyte model. To test the ability of the LRM to generalize, Fig 1D and 1E compare the LRM predicted P(EB)s to P(EB)s computed using the myocyte model for test data consisting of 100 independent MMI sets. The LRM performs well on the test set ($R^2$ = 0.993 for the 22 sets within the transition domain, mean prediction error = 0.006 ± 0.013). Detailed modeling methods and performance metrics are described in Methods and S1 Text.

The LRM for P(EB) was formulated with a total of 10 features. Four of these are linear features: $[Ca^{2+}]_i^{ini}$, $[Ca^{2+}]_{SR}^{ini}$, $G_{k1\_sf}$, and $k_{RyR}{}^+\_sf$. In addition, the following 6 derived quadratic features were used: $([Ca^{2+}]_{SR}^{ini})^2$, $[Ca^{2+}]_{SR}^{ini}*k_{RyR}{}^+\_sf$, $k_{RyR}{}^+\_sf^2$, $G_{K1\_sf}^2$, $([Ca^{2+}]_i^{ini})^2$, and $G_{K1\_sf}*k_{RyR}{}^+\_sf$. The selection of quadratic features is based on minimization of the Akaike information criterion (CAIC) [15] associated with this set of features (see Methods for details). Prior to fitting the LRM, features were scaled to a range of 0 to 1 in order to ensure interpretability of the LRM weights. Table 1 shows the LRM weights ranked by feature importance. Feature $[Ca^{2+}]_{SR}^{ini}$ is most important and is followed by feature $k_{RyR}{}^+$. $([Ca^{2+}]_{SR}^{ini})^2$ ranks as the third most important feature, indicating that quadratic features play an important role in improving model performance. In S2B Fig, the performance of the LRM built without quadratic features on the same test set has lower $R^2$ (0.862) in the transition domain and higher error (0.023 ± 0.058) than the 10-feature model shown in S2D Fig. This confirms that the additional 6 quadratic features improve LRM performance.

**Table 1. Ectopic beat features ranked by relative importance.**

| Ranking | Features | Weights ± SE from LRM |
|---------|----------|----------------------|
| 1 | $[Ca^{2+}]_{SR}{}^{ini}$ | 71.27 ± 3.81 |
| 2 | $k_{RyR}{}^{+}\_sf$ | 44.37 ± 2.56 |
| 3 | $([Ca^{2+}]_{SR}{}^{ini})^2$ | -24.41 ± 2.13 |
| 4 | $[Ca^{2+}]_{SR}{}^{ini*}k_{RyR}{}^{+}\_sf$ | -15.36 ± 2.76 |
| 5 | $k_{RyR}{}^{+}\_sf^{\,2}$ | -13.45 ± 1.14 |
| 6 | $[Ca^{2+}]_i$ | 12.77 ± 0.53 |
| 7 | $G_{K1}\_sf$ | -7.98 ± 0.62 |
| 8 | $G_{K1}\_sf^{\,2}$ | -3.62 ± 0.49 |
| 9 | $([Ca^{2+}]_i{}^{ini})^{\,2}$ | -2.68 ± 0.46 |
| 10 | $G_{K1}\_sf*k_{RyR}{}^{+}\_sf$ | -2.48 ± 0.54 |

These results presented here demonstrate that LRMs can accurately predict P(EB) generated by the complex myocyte model. For the 100 MMI test sets used in each case, thousands of CPU hours are required to obtain event probabilities from the myocyte model whereas evaluation of LRMs requires negligible computing time on the order of milliseconds. The computationally demanding task of estimating these probabilities is made possible by this LRM approach.

In Fig 2A–2D, P(EB) increases with increasing $[Ca^{2+}]_i{}^{ini}$ or $[Ca^{2+}]_{SR}{}^{ini}$. $I_{K1}$ downregulation also increases P(EB). Note that the P(EB) shown in Fig 2A–2D is predicted directly from the LRM. These findings agree with earlier simulations by Walker et al. [5]. The positive

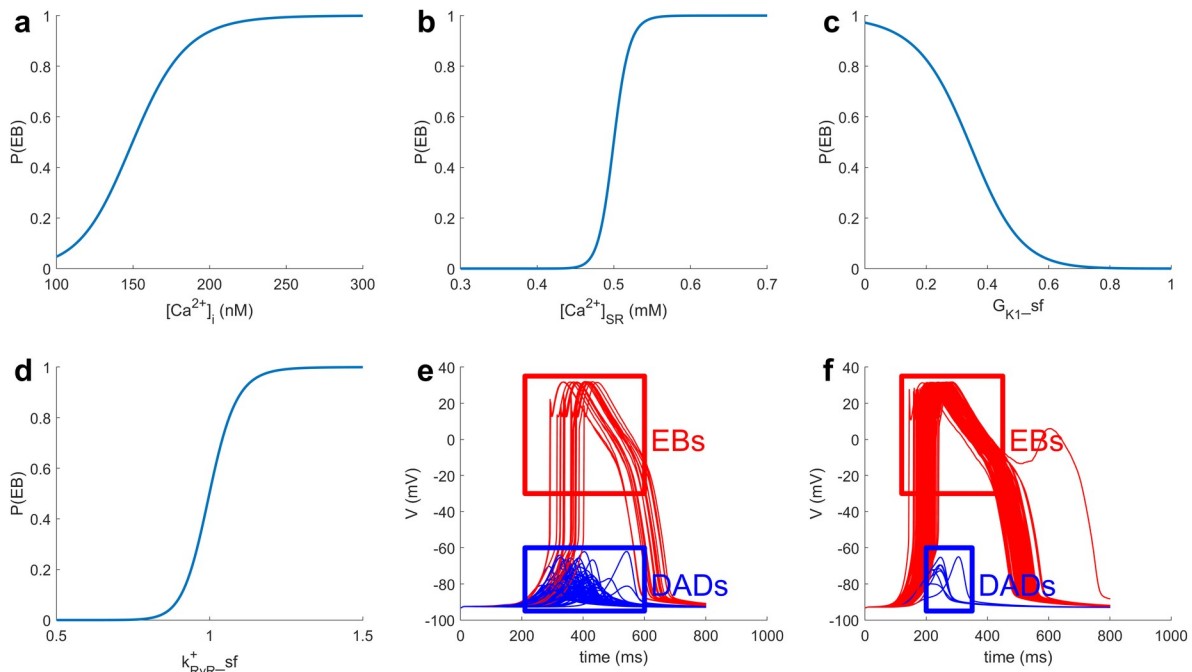

**Fig 2. Relationship between MMIs and P(EB).** In A-D, with exception of the MMI that is varied along the x-axis, MMIs are fixed at: $[Ca^{2+}]_i{}^{ini}$ = 150 nM, $[Ca^{2+}]_{SR}{}^{ini}$ = 0.5 mM, $G_{K1}\_sf$ = 0.338, and $k_{RyR}{}^{+}\_sf$ = 1. (E) 100 random myocyte model simulations with a low P(EB) = 0.19; MMI set: $[Ca^{2+}]_i{}^{ini}$ = 216 nM, $[Ca^{2+}]_{SR}{}^{ini}$ = 0.47 mM, $G_{K1}\_sf$ = 0.699, and $k_{RyR}{}^{+}\_sf$ = 1.12. (F) 100 random myocyte model simulations with a high P(EB) = 0.92; MMI set: $[Ca^{2+}]_i{}^{ini}$ = 154 nM, $[Ca^{2+}]_{SR}{}^{ini}$ = 0.60 mM, $G_{K1}\_sf$ = 0.775, and $k_{RyR}{}^{+}\_sf$ = 1.03. Red curves represent EBs and blue curves represent DADs that do not trigger ectopic beats.

correlation between $k_{RyR}^+$_sf and P(EB) agrees with the finding of Paavola et al. that $k_{RyR}^+$ is positively related to DAD induction [3]. Fig 2E and 2F show multiple myocyte model realizations for two example MMI sets. Fig 2E is an example with relatively low P(EB) obtained using the set: $[Ca^{2+}]_i^{ini}$ = 216 nM, $[Ca^{2+}]_{SR}^{ini}$ = 0.47 mM, $G_{K1}$_sf = 0.699, and $k_{RyR}^+$_sf = 1.12, where 19 out of 100 simulations generated ectopic beats. Fig 2F is an example with relatively high P(EB) obtained using the set: $[Ca^{2+}]_i^{ini}$ = 154 nM, $[Ca^{2+}]_{SR}^{ini}$ = 0.60 mM, $G_{K1}$_sf = 0.775, and $k_{RyR}^+$_sf = 1.03, where 92 out 100 simulations generated ectopic beats. The only difference across simulations in either Fig 2E or 2F is the random gating of LCCs and RyRs.

## Uncertainty analysis of P(EB)

The LRM allows for analysis of the uncertainty of probabilistic events such as ectopic beats arising from the experimental uncertainties inherent in measurements of the MMIs. We treat MMIs as random variables, and then the LRM becomes a probability transformation function (i.e. S3 Eq) of these MMI random variables. MMIs with uncertainties are assumed to have a normal distribution with σ taking the experimentally measured value for each MMI. From the experimentally derived distribution for each MMI, we randomly generate $10^6$ sets of MMIs and evaluate the LRM for each set to generate a predicted distribution of P(EB).

In the following study, the effect of $[Ca^{2+}]_{SR}^{ini}$ uncertainty on P(EB) was determined. In Fig 3A–3C, MMI set 1 was assumed to represent mean values for each MMI: $[Ca^{2+}]_i^{ini}$ = 150nM, $[Ca^{2+}]_{SR}^{ini}$ = 500μM, $k_{RyR}^+$_sf = 1, and $G_{K1}$_sf = 0.338, which yields P(EB) = 0.5. We assumed

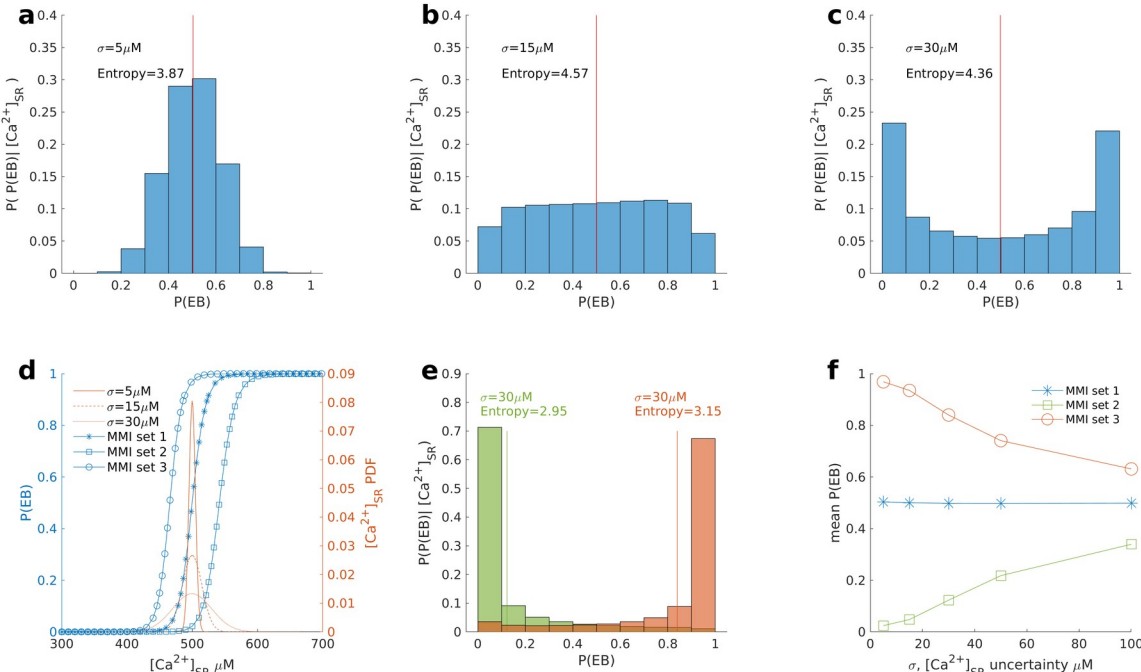

**Fig 3. Uncertainty analysis of P(EB) due to variation in $[Ca^{2+}]_{SR}^{ini}$.** Three MMI sets are used. All MMI sets have $[Ca^{2+}]_i^{ini}$ = 150 nM, $[Ca^{2+}]_{SR}^{ini}$ = 500 μM, and $k_{RyR}^+$_sf = 1. $G_{K1}$_sf for set 1, set 2, and set 3 is 0.338, 0.638, and 0.000, respectively. The P(EB) distributions given different hypothetical $[Ca^{2+}]_{SR}^{ini}$ uncertainties (σ) of set 1, where (A) σ = 5 μM, (B) σ = 15 μM, and (C) σ = 30 μM. The vertical red line is the mean of the P(EB) distribution. P(P(EB)|$[Ca^{2+}]_{SR}^{ini}$), on the y-axis, refers to the P(EB) distribution given $[Ca^{2+}]_{SR}^{ini}$ uncertainty. (D) Red curves are the $[Ca^{2+}]_{SR}^{ini}$ PDFs for σ = 5μM (solid line), σ = 15 μM (dashed line), and σ = 30 μM (dotted line). Blue curves are the model characteristic (MC) curves relating $[Ca^{2+}]_{SR}^{ini}$ and P(EB) for set 1 (stars), set 2 (open squares), and set 3 (open circles). (E) P(EB) distributions with $[Ca^{2+}]_{SR}^{ini}$ uncertainty σ = 30 μM for set 2 (green bars) and set 3 (red bars). (F) Relationship between mean P(EB) and $[Ca^{2+}]_{SR}^{ini}$ uncertainty for all three sets.

three different degrees of uncertainty in $[Ca^{2+}]_{SR}^{ini}$ corresponding to σ values of 5 μM, 15 μM, and 30 μM. In Fig 3A–3C, histograms show the P(EB) distribution given different $[Ca^{2+}]_{SR}^{ini}$ uncertainties, which leads to 3 different distribution patterns: unimodal (Fig 3A), approximately uniform (Fig 3B), and bimodal (Fig 3C). The red line indicates the mean P(EB). Fig 3A (σ = 5 μM) is a unimodal distribution in which P(EB) takes on values at or near 0.5 with high probability. Fig 3B (σ = 15 μM) is an approximately uniform distribution between 0 and 1, which indicates a complete lack of certainty in the value of P(EB).To quantify the degree of uncertainty in P(EB), entropy, a measure of the uncertainty of random variables, was evaluated for each P(EB) distribution shown in Fig 3 [3]. Entropy is more appropriate than the variance to assess uncertainty for multimodal distributions [16]. A detailed description of the entropy calculation is provided in S1 Text. The distribution of P(EB) in Fig 3B has maximum entropy. In Fig 3C (σ = 30 μM), the distribution of P(EB) becomes bimodal, with two peaks located at P(EB) values close to 0 and 1. In this scenario, the bimodal distribution indicates that P(EB) is either close to 0 or close to 1 given the symmetry of the P(EB) distribution. Thus, the entropy of the P(EB) distribution in Fig 3C less than that of Fig 3B.

In Fig 3D, blue curves show the $[Ca^{2+}]_{SR}^{ini}$-P(EB) relationship determined by the LRM, which we refer to as the model characteristic (MC) curve for $[Ca^{2+}]_{SR}^{ini}$, and the red curves show $[Ca^{2+}]_{SR}^{ini}$ probability density functions ($[Ca^{2+}]_{SR}^{ini}$ PDFs) which widen as $[Ca^{2+}]_{SR}^{ini}$ uncertainty σ increases. A comparison of the relative positions of the MC curve and the $[Ca^{2+}]_{SR}^{ini}$ PDF can be used to explain the previously described P(EB) distribution patterns. The MC curve for set 1 (stars) shows the $[Ca^{2+}]_{SR}^{ini}$-P(EB) relationship as $[Ca^{2+}]_{SR}^{ini}$ is varied. The width of the $[Ca^{2+}]_{SR}^{ini}$ PDF for σ = 5 μM (solid curve) is quite narrow and exists at the center of the transition domain where P(EB) = 0.5. As a result, the distribution of P(EB) is unimodal with peak near 0.5. The width of the $[Ca^{2+}]_{SR}^{ini}$ PDF for σ = 15 μM (dashed curve) coincides with the boundaries of the transition domain such that the distribution of P(EB) in Fig 3B is approximately uniform. The $[Ca^{2+}]_{SR}^{ini}$ PDF for σ = 30μM (dotted line) is much wider such that a sufficiently large portion of the $[Ca^{2+}]_{SR}^{ini}$ PDF coincides with the lower plateau domain and the upper plateau domain of the MMI set 1 MC curve, where P(EB) takes on values of either 0 or 1 resulting in a bimodal distribution. This behavior reveals an interesting finding that with increasing uncertainty of $[Ca^{2+}]_{SR}^{ini}$, uncertainty in P(EB) first increases and then decreases. To confirm our finding, we calculated the entropy of P(EB) distributions for Fig 3A–3C by S4 Eq. The P(EB) distribution with σ = 15 μM has the highest entropy which validates our finding. The mean P(EB)s for all distributions shown in Fig 3A–3C are 0.5 because all variations in $[Ca^{2+}]_{SR}^{ini}$ occur about a mean $[Ca^{2+}]_{SR}^{ini}$ = 500 μM in this scenario.

The role of $[Ca^{2+}]_{SR}^{ini}$ uncertainty on P(EB) is also evaluated for two additional sets for which P(EB) ≠ 0.5: set 2 ($[Ca^{2+}]_i^{ini}$ = 150 nM, $[Ca^{2+}]_{SR}^{ini}$ = 500 μM, $k_{RyR}^+$_sf = 1, and $G_{K1}$_sf = 0.638, for which P(EB) = 0.02) and set 3 ($[Ca^{2+}]_i^{ini}$ = 150 nM, $[Ca^{2+}]_{SR}^{ini}$ = 500 μM, $k_{RyR}^+$_sf = 1, and $G_{K1}$_sf = 0.0, for which P(EB) = 0.97). The distribution of P(EB) for sets 2 and 3 are evaluated for $[Ca^{2+}]_{SR}^{ini}$ uncertainty with σ = 30μM. In Fig 3E, the P(EB) distribution of $[Ca^{2+}]_{SR}^{ini}$ is unimodal for both set 2 and set 3. MC curves for both set 2 (open squares) and 3 (open circles) are shown in Fig 3D. For set 2, the $[Ca^{2+}]_{SR}^{ini}$ PDF coincides with the lower plateau domain and part of the transition domain of the MC curve. This results in a P(EB) distribution that is unimodal with P(EB) values close to 0. In a similar manner, the $[Ca^{2+}]_{SR}^{ini}$ PDF for set 3 primarily coincides with the upper plateau domain and yields a P(EB) distribution that is a unimodal with P(EB) values close to 1. These demonstrate that the P(EB) distribution pattern depends on the width of $[Ca^{2+}]_{SR}^{ini}$ PDF (degree of $[Ca^{2+}]_{SR}^{ini}$ uncertainty) as well as the relative position of the mean $[Ca^{2+}]_{SR}^{ini}$ with respect to the transition domain of its MC curve.

In general, $[Ca^{2+}]_{SR}^{ini}$ uncertainty impacts the mean of the P(EB) distribution. Fig 3F shows the relationship of $[Ca^{2+}]_{SR}^{ini}$ uncertainty (σ) and the mean P(EB) for sets 1–3. Experimental

uncertainty in measurements of $[Ca^{2+}]_{SR}^{ini}$ varies widely from 10 µM to 290 µM [17–19], and we therefore evaluated 0 µM $\leq \sigma \leq$ 100 µM. Results show that with the increase of $[Ca^{2+}]_{SR}^{ini}$ uncertainty, the mean P(EB) converges to 0.5 regardless of the selected set. When the $[Ca^{2+}]_{SR}^{ini}$ PDF is narrow (e.g., $\sigma$ = 5µM), the P(EB) distribution will be unimodal centered at the value of P(EB) defined by the LRM for the given set, and therefore, the mean P(EB) will be close to this value. In contrast, when the $[Ca^{2+}]_{SR}^{ini}$ uncertainty is sufficiently large, a majority of the $[Ca^{2+}]_{SR}^{ini}$ PDF will coincide with the lower plateau domain and upper plateau domain of the MC curve, with only a small to negligible portion of the $[Ca^{2+}]_{SR}^{ini}$ PDF coinciding with the transition domain of the MC curve. In the limit as $\sigma$ grows to be large, for any set, the distribution of $[Ca^{2+}]_{SR}^{ini}$ flattens such that there is equal probability of it taking on values in either the lower plateau domain or upper plateau domain of the MC curve, and hence P(EB) becomes equally likely to take on values of 0 or 1 and mean P(EB) converges to 0.5 (Fig 3F).

## Role of $G_{K1}$ on P(EB)

The effect of $I_{K1}$ current density scaling factor ($G_{K1\_sf}$) uncertainty on P(EB) was evaluated. We used the experimental results of Pogwizd et. al. [20] to obtain the distribution of $G_{K1\_sf}$ values in both normal and heart failure (HF) myocytes. The process by which the LRM parameters (i.e. weights) are constrained is described in detail in the supplement (S3A and S3B Fig). The following three MMIs are fixed as follows: $[Ca^{2+}]_i^{ini}$ = 150 nM, $[Ca^{2+}]_{SR}^{ini}$ = 550 µM, and $k_{RyR}^{+}\_sf$ = 1. $G_{K1\_sf}$ is treated as uncertain where in normal cells it is 1.0000 ± 0.1108 ($\sigma_N$) and in heart failure (HF) cells it is 0.5100 ± 0.0852 ($\sigma_{HF}$). The P(EB) corresponding to the mean value of $G_{K1\_sf}$ is 0.008 and 0.918 for normal and HF, respectively. Fig 4A shows the P(EB) distributions arising from the experimental uncertainty of $G_{K1\_sf}$. The P(EB) arising from the experimental distribution of $G_{K1\_sf}$ is 0.0294 ± 0.0611 in normal cells and 0.8788 ± 0.1183 in HF. Results show that, even when accounting for experimental uncertainty, P(EB) for HF $G_{K1\_sf}$ is significantly higher than that corresponding to normal $G_{K1\_sf}$. Fig 4B shows the relative positions of the $G_{K1\_sf}$ MC curve (blue) and the $G_{K1\_sf}$ PDFs (red), which underly the P(EB) distributions in Fig 4A.

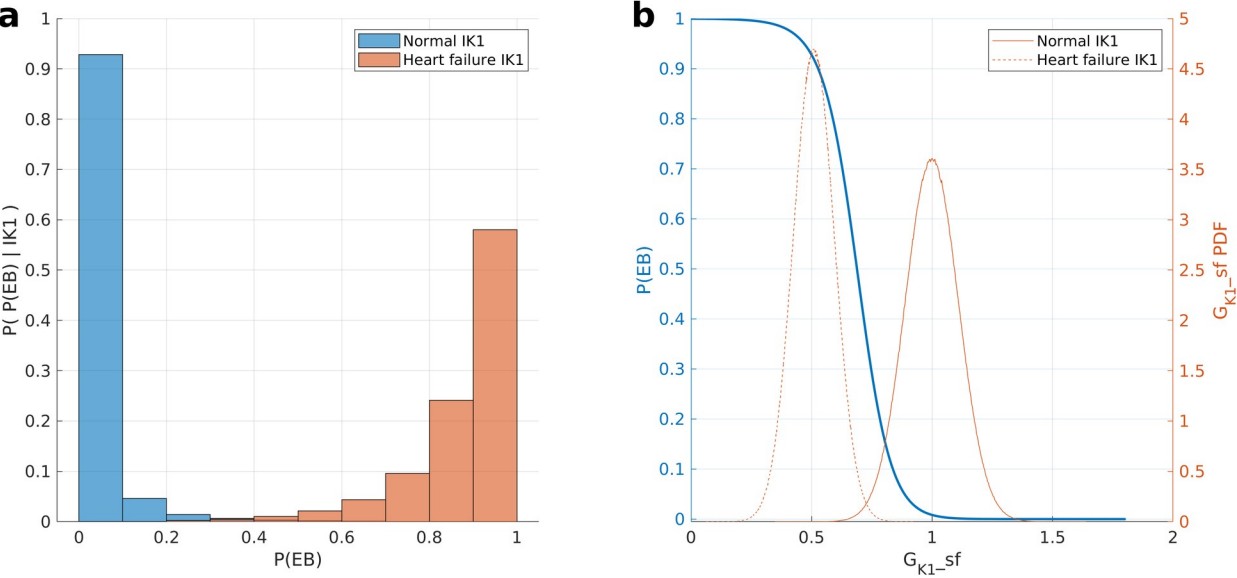

**Fig 4. Effect of $G_{K1\_sf}$ uncertainty on P(EB).** (A) Distributions of P(EB) for normal $G_{K1\_sf}$ (blue) and HF $G_{K1\_sf}$ (red). (B). MC curve (blue) shows the relationship of $G_{K1\_sf}$ and P(EB) for the LRM. $G_{K1\_sf}$ PDFs for normal (solid red line) and HF (dashed red line).

## Discussion

Regression models have been used previously by Sobie et al. to estimate action potential duration, peak and resting membrane potential, and $Ca^{2+}$ transient amplitude from deterministic myocyte models, as well as the probability of the occurrence of $Ca^{2+}$ sparks from a stochastic $Ca^{2+}$ release site model [6,7]. Based on their method, we developed a new pipeline for establishing simplified models (LRMs) for estimating the probability of cellular cardiac arrhythmia events simulated by a complex stochastic myocyte model. Fig 1E shows excellent prediction performance of the LRM in predicting the probability of ectopic beats. The method developed in this work is universal in the sense that it should be considered a general approach to simplifying highly complex nonlinear stochastic models in a wide range of applications.

Our approach differs from that of Lee at al. [7] at two points in the modeling pipeline. First, we developed a two-iteration strategy for generating MMI sets. The second iteration ensures that the transition domain is adequately sampled. Second, we derived LRM quadratic features from the MMIs. The rationale for this approach is that since the myocyte model is highly nonlinear, we expect improved performance by allowing for a nonlinear relationship between MMIs and P(EB) in the LRM. To demonstrate the performance improvement of these two steps, we built preliminary LRMs from only the first iteration MMI sets and using only MMIs as features (i.e. without derived quadratic features). For prediction of P(EB), this approach yields an average error of 0.026 ± 0.058 (S2A Fig), which is worse than that (0.006 ± 0.013) obtained with the full pipeline (S2D Fig).

The LRM approach has the benefit of computational efficiency. The average computational time required for each MMI set in the EB study is ~0.4 (hours per realization) * 100 (realizations) = 40 CPU hours. To build the LRM, however, 200 training MMI sets must be simulated with the myocyte model, which takes ~8000 CPU hours. After building the LRM, calculating for new MMI sets takes negligible time. For example, the computational time required to run the test set (100 MMI sets) in the ectopic beat study is therefore 4000 CPU hours. In contrast, the computational time for LRMs is negligible (~5 CPU ms). Additionally, in the uncertainty study of $[Ca^{2+}]_{SR}^{ini}$, $10^6$ samples were randomly sampled from the uncertainty range of $[Ca^{2+}]_{SR}^{ini}$. If we estimate the P(EB) for each sample, the total computational time required is $0.4 * 100 * 10^6 = 4*10^7$ CPU hour. In contrast, LRM takes less than 1 CPU second. These results show the dramatic improvement in computational efficiency of the LRM.

LRMs enable us to quantitatively explore the relationship between MMIs and event probabilities. In the ectopic beat study, Fig 2A and 2B show that P(EB) increases with increasing $[Ca^{2+}]_i^{ini}$ and $[Ca^{2+}]_{SR}^{ini}$, indicating that elevation of cellular $Ca^{2+}$ (i.e. $Ca^{2+}$ overload) tends to increase the likelihood of ectopic beats. This result is consistent with previous experimental and computational studies [5,21]. Fig 2C shows that increasing $G_{K1}\_sf$ reduces P(EB). $I_{K1}$ downregulation has been shown to be associated with ventricular arrhythmias such as Anderson's syndrome [22] and LQTS [23]. Increased $k_{RyR}^+\_sf$ is also shown in Fig 2D to elevate P(EB). Kubalova et al. showed that heart failure myocytes exhibit elevated RyR open probability [24]. Paavola et. al. also demonstrated the positive correlation between RyR opening rate with DAD induction and $Ca^{2+}$ waves [3]. Table 1 demonstrates the importance ranking of features in the LRM for P(EB). Feature $[Ca^{2+}]_{SR}^{ini}$ is the most important among all 10 features incorporated in the LRM. Additional MMIs that were not considered in this study may also have a significant impact on P(EB). Thus, feature $[Ca^{2+}]_{SR}^{ini}$ is not necessarily the most important feature among all possible myocyte model inputs.

To further analyze the importance of features in P(EB) prediction, we trained 9 sub-models which included only the top $n$ ($n$ ranging from 1 to 9) features showed in Table 1 on the same 200 training MMI sets, respectively. The $R^2$ and error (mean ± SD) were then obtained for all

9 sub-models on the same 100 test MMI sets (S1 Table). The sub-model based on the top 7 features ($n = 7$) has mean error of P(EB) = 0.013 and $R^2$ = 0.995, whereas the $n = 6$ sub-model has mean error P(EB) = 0.119 and $R^2$ = 0.758. This indicates that the $n = 7$ sub-model is the smallest sub-model with performance similar to the full model ($R^2$ = 0.999, mean error = 0.006). In addition, the $n = 7$ sub-model is the smallest sub-model containing all four MMIs. These results indicate that all 4 MMIs evaluated in this study are necessary to achieve a high level of LRM performance regardless of their relative importance rankings.

In Fig 3A–3C, in which $[Ca^{2+}]_{SR}^{ini}$ variation was explored, three P(EB) distribution patterns (unimodal, approximately uniform, and bimodal) were revealed when exploring the role of $[Ca^{2+}]_{SR}^{ini}$ uncertainty on predictions of P(EB). This implies that with the increase of $[Ca^{2+}]_{SR}^{ini}$, the uncertainty of P(EB) first increases and then decreases. This result is also confirmed by comparing the entropies of P(EB) distributions associated with different degrees of uncertainty in $[Ca^{2+}]_{SR}^{ini}$. Reported experimental uncertainties (standard errors) of $[Ca^{2+}]_{SR}^{ini}$ vary widely from 10μM to 290μM [17–19]. A consequence of this is that for the majority of the realistic uncertainty range (30–290 μM), myocytes are unlikely to be in the transition domain where both DADs and EBs occur with nonzero probability. Namely, with a small perturbation in $[Ca^{2+}]_{SR}^{ini}$ relative to the uncertainty range, the most likely behavior of the cell would jump between always exhibiting DADs and always exhibiting EBs. This is consistent with the fact that the EB is a bifurcation phenomenon. Other studies also showed that EBs arise as a result of a bifurcation [25].

In Fig 4A, the heart failure (HF) $G_{K1}$ study shows that, given the experimental uncertainty, P(EB) associated with HF $G_{K1}$ is statistically significantly higher than P(EB) of normal $G_{K1}$. Consistent with our results, Maruyama et al. showed that suppressing $I_{K1}$ considerably enhanced the DAD amplitude and P(EB) [26]. This result shows the potential in using the LRM-predicted probability of an arrhythmic event to test the propensity for arrhythmias resulting from variations in other ion currents. We chose to explore $G_{K1}$ in HF myocytes because $I_{K1}$ data were available, and this method can be used to explore the impact of variation in other MMIs as well. These results also demonstrate the potential role for P(EB) as a useful biomarker for arrhythmia predictions, particularly in mutations and diseases linked to DADs such as catecholaminergic polymorphic ventricular tachycardia [27]. Additionally, early afterdepolarization (EAD), another important type of cellular arrhythmia, can be predicted via LRM. Such an LRM may be used as a biomarker to predict the arrhythmogenic risk of EAD-related diseases such as long QT syndrome [28].

As shown by two case studies (uncertainty of $[Ca^{2+}]_{SR}^{ini}$ as well as $G_{K1}$ in heart failure), The LRM-based uncertainty analysis approach presented in this study enables efficient propagation of experimentally measured uncertainty to P(EB) prediction. To further demonstrate the broad significance of this uncertainty analysis approach, two additional potential applications are discussed. 1) In this work, each application of uncertainty analysis has explored the impact of uncertainty in a single MMI ($[Ca^{2+}]_{SR}^{ini}$ or $G_{K1}$). However, this uncertainty analysis approach can be applied to evaluate the role of uncertainties of all MMIs simultaneously. Such an approach may better mimic real experimental conditions where multiple quantities are being measured or controlled. 2) Additionally, this uncertainty approach can serve as a guide to experimentalists for the determination of the degree of accuracy required in any particular measurement in order to yield a reasonable prediction of P(EB).

A potential future application of our P(EB) prediction approach will be to systematically study the relationship between P(EB) and the delayed afterdepolarization. It is well understood that triggered ectopic beats arise from DADs when DAD amplitude is sufficiently large to activate rapid inward $Na^+$ current [2]. Liu et al. have known that the DAD amplitude threshold for generating ectopic beats is dynamic [29]. These results imply that the relationship between

P(EB) and DAD amplitude is not necessarily a simple monotonically increasing function. Some interesting as yet unanswered questions could be explored. For example, under what conditions (i.e., what set of MMIs), might P(EB) be relatively large while DAD amplitude distribution tends to smaller values, and vice versa. How does the uncertainty in DAD amplitude relate to the uncertainty in P(EB)s?

In summary, we developed a pipeline for building logistic regression models (LRMs) that predict the probability of cellular arrhythmias. These simplified models faithfully reproduce the relationship between parameters/inputs and probabilistic events as learned from the mechanistic biophysically-detailed stochastic myocyte models, but with far less computational burden. As far as we know, this is the first simplified model enabling quantitative investigation of determinants of the probability of cellular arrhythmia events in a complex stochastic spatial myocyte model. This approach, which we refer to as arrhythmia sensitivity analysis, allows for systematic study of the relationship between these event probabilities and their associated myocyte model inputs. As an example, we built a LRM to study the relationship between P(EB) and 4 MMIs: $[Ca^{2+}]_i^{ini}$, $[Ca^{2+}]_{SR}^{ini}$, $G_{K1}$ and $k_{RyR}^+$. The negligible computational burden of LRM allows us to perform uncertainty analysis. In the $[Ca^{2+}]_{SR}^{ini}$ uncertainty study, we showed and rationalized an emergent property of the P(EB) distribution, where with increasing $[Ca^{2+}]_{SR}^{ini}$ uncertainty, P(EB) uncertainty first increases and then decreases. Lastly, the heart failure $G_{K1}$ study shows the potential value of our approach in revealing the determinants of arrhythmia probability, indicating the potential for this method to be used as a tool for arrhythmia risk prediction.

## Methods

### Modified walker ventricular myocyte model

This study uses a modified version of the Walker et al. stochastic ventricular myocyte model [5]. The Walker et al. model is a biophysical-detailed three-dimensional spatial stochastic model of a cardiac ventricular myocyte containing 25,000 $Ca^{2+}$ release units (CRUs) (each representing a dyad) along with their sub-membrane compartments. A subset of CRUs (5400) was used for all simulations to accelerate the computation and all CRU $Ca^{2+}$ fluxes were scaled to the total number of CRUs. This model has been shown to reproduce realistic $Ca^{2+}$ waves as well as DADs. Chu et al. [12] recently developed a model of the $Na^+$-$Ca^{2+}$ exchanger (NCX) that includes allosteric regulation of NCX function by $Ca^{2+}$. This NCX model was incorporated into the Walker et al. model. Following Chu et al. [12], NCX was localized 15% in dyad, 30% in the submembrane compartment, and 55% in the cytosol. In addition to these changes, NCX conductance was increased 20%, the time constant of $Ca^{2+}$ flux from submembrane to cytosol was reduced by 20%, $I_{Kr}$ conductance increased by 20%, SERCA pump rate decreased by 25%, and number of $I_{to2}$ channels reduced from 8 to 5 in each submembrane compartment to assure a more realistic canine APD (~290 ms) and $Ca^{2+}$ transient peak (~800 nM), similar to that of the normal canine ventricular myocyte [13].

### Simulation protocol for ectopic beat

In the ectopic beat studies, the initial state of the myocyte model corresponds to the diastolic state, where membrane potential is -93.4 mV and NCX is partially allosterically activated (~20%-40%). To create the condition that favors the generation of DADs and ectopic beats, model simulations are performed under a β-adrenergic stimulation protocol, which is the same as previously described by Walker et al. [5,24], with the exception that RyR opening rate is increased 4-fold compared to its control value based on experimental data for heart failure

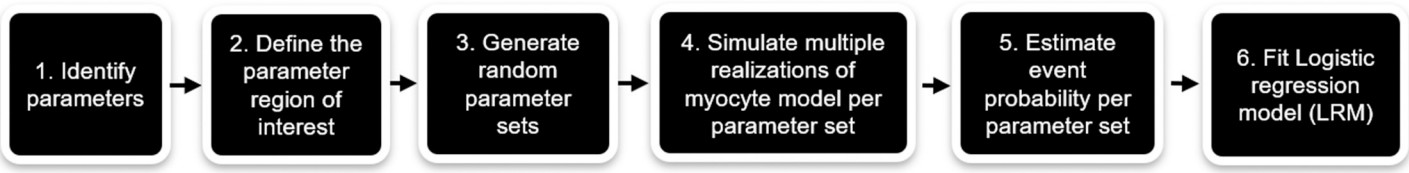

**Fig 5. General workflow pipeline for logistic model formulation.**

[14] and to increase the propensity for EBs. No external current was applied, and simulation duration is up to 800ms. Please see details in S1 Text.

## Statistical modeling

Fig 5 illustrates the general workflow for building a logistic regression model (LRM) from an underlying biophysically-detailed myocyte model. A more detailed workflow (S1 Fig) and its mathematical development is provided in the supplement. The steps of the workflow in Fig 5 are performed twice to build the LRM. In the first iteration, relevant myocyte model inputs (MMIs), which refers collectively to myocyte model parameters as well as state variable initial conditions, and their ranges of variation (regions of interest) are identified. A region of interest is established to keep MMIs within biophysically relevant ranges. A detailed description of region of interest is in the next section. Multiple randomly selected MMI sets are generated by uniform sampling within this region of interest. For each sampled MMI set, multiple simulations (realizations) are performed using the modified Walker model. A binary indicator random variable is defined using the output of each simulation. This random variable indicates whether the cellular arrhythmia event of interest did or did not occur. For each MMI set, an event (occurrence of an ectopic beat) probability is estimated based on its frequency of occurrence over all realizations. All MMI sets and their corresponding event probabilities are then used to derive a logistic regression model relating the probability of the event to a weighted sum of MMIs:

$$P(\text{event}) = \frac{1}{1 + \exp(-\underline{B}^\mathrm{T}\underline{P})} \tag{1}$$

, where $\underline{P} = \underline{P}_o$ is a vector of features, $\underline{B} = \underline{B}_o$ is a vector of weights. $\underline{P}_o$ consists only of linear features (i.e. MMIs). The subscript "$o$" stands for "initial", representing the first iteration of the pipeline. P(event) represents the estimated event occurrence probability.

After this first iteration, the logistic equation Eq 1 is then used as an additional constraint to estimate the transition domain, which is a subregion within the region of interest for the MMI set. A detailed description of transition domain is in the next section. The second iteration then samples only within the transition domain, and steps 4–5 of the workflow are repeated. MMI sets and corresponding event probabilities obtained from both iterations are combined, and the final LRM is built again using Eq 1, where $\underline{P} = \underline{P}_f$ is the final feature vector and $\underline{B} = \underline{B}_f$ is the final weights vector. The subscript "$f$" stands for "final", representing the second iteration of the pipeline. In addition to linear features (MMIs), $\underline{P}_f$ also includes quadratic features derived from the linear features. To select the best performing combination of quadratic features for $\underline{P}_f$, we formulated the following approach. We first construct a set of feature vectors which includes all possible combinations of quadratic features. Each feature vector in this set consists of all linear features and a unique combination of quadratic features. For each feature vector, we fit Eq 1 with the MMI sets from first and second iterations. We choose, as the best

performing set of quadratic features, those that comprise the particular feature vector which minimizes the Akaike information criterion (CAIC) [15].

We refer to this model as the LRM. 100 training MMI sets are generated in both the first and second iterations. For each MMI set, 100 realizations (for ectopic beat study) are simulated to estimate the probability of the event of interest. To ensure the generalizability of the LRM, an additional independent 100 test MMI sets are generated within the region of interest to be used for evaluation of the performance of the LRM. All logistic regression model fits were performed with MATLAB R2018b *fitglm* function.

## MMI identification and MMI region of interest

When building the LRM for predicting probability of occurrence of DAD-induced EBs (P(EB)), four MMIs were selected to be varied. These were initial states of $[Ca^{2+}]_i$ and $[Ca^{2+}]_{SR}$, and scaling factors controlling $I_{K1}$ current density ($G_{K1}\_sf$) and RyR opening rate ($k_{RyR}^+\_sf$). $[Ca^{2+}]_i^{ini}$ and $[Ca^{2+}]_{SR}^{ini}$ are defined as the initial state values of $[Ca^{2+}]_i$ and $[Ca^{2+}]_{SR}$. A region of interest is selected for each MMI. Shannon et al. showed that the diastolic total SR $[Ca^{2+}]$ load varies between 58–134 μM based on different pacing frequencies [18]. Thus, the corresponding range of free intra-SR $[Ca^{2+}]$ is 224 – 900mM, and we therefore selected the range of 300–700μM for $[Ca^{2+}]_{SR}^{ini}$. Beuckelmann et al. showed that the resting $[Ca^{2+}]_i$ is 96 ± 47nM for normal myocytes and 165 ± 61 nM for heart failure myocytes [30]. Thus, we selected the range of 100 – 300nM for $[Ca^{2+}]_i^{ini}$. Down-regulation of $I_{K1}$ [20] has been shown to occur in failing heart myocytes and changes in these cellular properties are associated with cellular arrhythmias. Thus, the range of 0–1 is selected for $G_{K1}\_sf$. The range for $k_{RyR}^+\_sf$ is 0.5–1.5.

The default MMI set corresponds to $G_{K1}\_sf = 1$ and $k_{RyR}^+\_sf = 1$. $[Ca^{2+}]_{SR}^{ini}$ and $[Ca^{2+}]_i^{ini}$ are selected to match diastolic levels while under β-adrenergic stimulation (based on steady state 0.5Hz pacing). $[Ca^{2+}]_{SR}^{ini}$ is set as 700mM, derived from the experimental total SR $Ca^{2+}$ load of 111 μM [31]. The diastolic $[Ca^{2+}]_i$ under β-adrenergic stimulation is consistent with the normal condition such that $[Ca^{2+}]_i^{ini}$ is set to 100nM [32]. P(EB) for this default parameter set is 0.76 predicted from the LRM and 0.74 obtained by 100 realizations performed on the myocyte model. These baseline values indicate that model simulations are performed under conditions that favor of the generation of DADs and ectopic beats.

The region of interest can be separated into two mutually exclusive subspaces in terms of the arrhythmia event probability: plateau domain and transition domain. The plateau domain is the space where MMI sets simulated in the myocyte model yield P(event) strictly equal to 0 or 1. The transition domain is the space where MMI sets simulated in the myocyte model yield P(event) strictly > 0 and < 1. We can specify the lower plateau domain as corresponding to P(event) = 0 and the upper plateau domain as corresponding to P(event) = 1 since these are also mutually exclusive. Because of the finite number of realizations performed with the myocyte model for each MMI set, P(event) takes on a discrete set of values such that we can strictly follow this definition.

## LRM validation metrics

Linear regression was performed between values of myocyte model-generated actual P(event) and LRM predicted P(event). The R-squared ($R^2$) from the linear regression was calculated. Absolute error was also calculated as

$$\text{Error} = \frac{\sum_{i=1}^{n} |P_i(event)_{MM} - P_i(event)_{LRM}|}{n} \tag{2}$$

,where $n$ is the number of MMI sets, $P_i(event)_{MM}$ is event probability of $i$th MMI set obtained

from the myocyte model, and $P_i(event)_{LRM}$ is event probability of $i$th MMI set obtained from the LRM.

## Uncertainty analyses

The LRM derived in this study is a function that maps MMIs to P(event). Experimental measurements of these MMIs exhibit variability, and this variation can be modeled by assuming these measurements are drawn from an underlying probability distribution. In this case, the event probability produced by the LRM is itself a random variable. We wish to characterize this uncertainty by computing the distribution of the estimated event probability. To do this, when the mean and variance of MMI estimates are available from experimental data, we assume MMIs are random variables drawn from a normal distribution with that mean and variance. In cases where the argument of the LRM ($\underline{B}_f^T \underline{P}_f$) involves a weighted sum of individual MMIs not including quadratic features, this distribution can be calculated analytically, and is the logit-normal distribution (Eq. S5) [33]. In cases where the feature vector $\underline{P}_f$ includes elements containing quadratic features, the distribution of P(event) need to be estimated numerically. In this latter case, we randomly generate $10^6$ sample vectors for a specific MMI set based on the underlying MMI uncertainty distributions and use the LRM to predict P(event) for all sample vectors. Following this approach, the P(event) distribution can be estimated.

## Supporting information

**S1 Fig. Detailed workflow for building the 2-iteration logistic regression model (LRM).**
(TIF)

**S2 Fig. DAD-induced ectopic beat (EB) prediction performance for different LRMs.** Performance of each LRM is evaluated on the same test set which consists of 100 myocyte model input (MMI) sets for all models. Each LRM has a unique training strategy. A) LRM trained with only linear features (excluding quadratic features) and only 100 MMI sets from only the first iteration. B) LRM trained with only linear features (excluding quadratic features) and 200 MMI sets (100 sets in the first iteration + 100 transition domain sets in the second iteration). C) LRM trained with linear features and quadratic features on 100 MMI sets from only the first iteration. D) Reproduction of Fig 1E, LRM trained with linear features and quadratic features and 200 MMI sets (100 sets in the first iteration + 100 transition domain sets in the second iteration). Linear features and quadratic features used in A-D are given in Table 1. First iteration training data was identical for all LRMs and second iteration training data was identical for LRMs in panels B and D. $R^2$ for the entire region of interest and transition domain as well as the average error are reported.
(TIF)

**S3 Fig. Constraining $G_{K1}$ scaling factor ($G_{K1}\_sf$).** Experimental $I_{K1}$ I-V relationship for normal and Heart failure (HF) is measured and showed in Fig 5C of Pogwizd et. al. In the experimental protocol, $I_{K1}$ is measured in response to 500-ms steps from a holding potential of -30 mV to test potentials in the range of -120 mV to +40 mV in 6 normal cells and 6 HF cells. The detail of the protocol is described in Pogwizd et. al. Error bars represent standard error (SE). (A) Best fit of the standard deviation (SD) for the normal $I_{K1}$ using the same voltage clamp protocol. (B) Best fit of the SD for the HF $I_{K1}$ using the same voltage clamp protocol.
(TIF)

**S1 Table. Sub-model performance on the 100 MMI test set.**
(DOCX)

**S1 Text. Supporting description of logistic regression model and entropy.**
(DOCX)

## Author Contributions

**Conceptualization:** Qingchu Jin, Joseph L. Greenstein, Raimond L. Winslow.

**Formal analysis:** Qingchu Jin.

**Funding acquisition:** Raimond L. Winslow.

**Investigation:** Qingchu Jin, Joseph L. Greenstein, Raimond L. Winslow.

**Methodology:** Qingchu Jin, Joseph L. Greenstein, Raimond L. Winslow.

**Resources:** Raimond L. Winslow.

**Supervision:** Raimond L. Winslow.

**Visualization:** Qingchu Jin.

**Writing – original draft:** Qingchu Jin.

**Writing – review & editing:** Qingchu Jin, Joseph L. Greenstein, Raimond L. Winslow.

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
