## [Decision Letter · Decision Letter 0]

19 Apr 2021

Dear Dr. Winslow,

Thank you very much for submitting your manuscript "Estimating Ectopic Beat Probability with Simplified Statistical Models that Account for Experimental Uncertainty" for consideration at PLOS Computational Biology.

As with all papers reviewed by the journal, your manuscript was reviewed by members of the editorial board and by several independent reviewers. In light of the reviews (below this email), we would like to invite the resubmission of a significantly-revised version that takes into account the reviewers' comments.

We cannot make any decision about publication until we have seen the revised manuscript and your response to the reviewers' comments. Your revised manuscript is also likely to be sent to reviewers for further evaluation.

Sincerely,

Daniel A Beard

Deputy Editor

PLOS Computational Biology

Reviewer's Responses to Questions

**Comments to the Authors:**

Reviewer #1: The study by Jin et al is an interesting study that presents the utility of a statistical model (specifically a logistic regression model) to predict the probability of an ectopic beat in an isolated cardiac cell, based on a detailed biophysical model that incorporates stochastic calcium cycling. The study is of interest to the computational biology community and the manuscript is well-written. However, I have some concerns for the authors to address:

Major:

1. The rationale of the specific 10 features used in the LRM is unclear as presented in the main text, specifically why only certain quadratic terms are used, and if some feature selection process was chosen. Eventually I found this detail in the Supporting Methods – that the LRM was chosen based on CAIC – but the details of this process need to be included in the main text.

Regardless, why did the authors choose these specific 4 MMIs to consider for the LRM? Presumably the model is also highly dependent on other inputs, specifically Ca cycling related parameters. This seems to bias the analysis that could be missing other parameters equally or more important in EB triggers.

2. It seems like the analysis/simulations presented will also produce predictions for the probability of DADs, in addition to EBs. Can the authors compare the ability to use the LRM to predict P(DADs)? It would be valuable to compare conditions that lead to DADs with conditions that lead to EBs. Additionally, the LRM framework would enable particularly useful predictions. For example, how does uncertainty in DADs relate to uncertainty in EBs? Similarly, does uncertainty in DADs relate to changes in P(EB)? Does DADs uncertainty decrease as P(EB) increases?

3. It is interesting that the top 5 terms in the LRM by weight only contain terms for Ca_SR and kRyR_sf. Did the authors consider the sub-model that only contains these terms and compare with the full 10 term model? The relative weights suggests that this model may be quite good at prediction as well.

4. The author’s note that their work builds off of prior work from the Sobie lab, and it seems that prediction of an output distribution and uncertainty is one novel aspect of the authors’ extension. This could probably be highlighted to a greater extent.

Additionally, measurement of uncertainty could more broadly be applied to consider practical aspects of experimental measurements, by investigating parameter uncertainty in all inputs at once. For example, if there is parameter uncertainty in all inputs with a magnitude of 10%, how much uncertainty propagates to the P(EB) prediction? How much uncertainty is “tolerable” in measurements in order to make reasonable predictions of P(EB)?

Minor:

1. There are some terminology ambiguities regarding describing Ca_i and Ca_SR as parameters early in the paper. This is eventually clarified when the authors introduce the nomenclature of MMIs, which includes the initial conditions for these state variables, but this is unclear in the abstract and introduction. I would suggest defining a different notation for the initial conditions, e.g., Ca_i^0 and Ca_SR^0, to distinguish these values from the state variables.

2. In Figure 2, from context it seems that plots in panels A-D are calculated directly from the LRM, not simulations, but this is unclear. Please clarify in the text.

3. The methods describe an Eqn S5 in the Supplement, but this equation is not included in the Supplement.

4. This may be an issue with the journal manuscript submission site, but the quality of all of the figures is very poor and it is difficult to read the text in most of the figures.

Reviewer #2: Jin et al present a clever approach for simplifying a complex, non-linear, and stochastic mathematical model by using logistic regression for estimating the probability of spontaneous beats. This can provide computational advantages, and is a good way of summarizing which variables are important in the observance of a phenomenon.

This is a good application of these sorts of simplified statistical models, and there’s a lot to like in the manuscript. Nonetheless, there are also weaknesses that should be addressed.

(1) I initially didn’t understand the simulation protocol, and I wasn’t sure how SR [Ca] could be considered a parameter, since this is a variable that changes with time in contemporary myocyte models. On the initial reading I was envisioning a cell that was being paced for roughly 100 beats, then the authors were observing whether spontaneous Ca release was observed after pacing was stopped. So the setup needs to clarify a that these cells are not regularly paced, and that they’ve created conditions where spontaneous release in the form of Ca waves always occurs (or at least I assume it does). Although these conditions are somewhat artificial for healthy myocytes, I can still see the value in the modeling.

(2) The text is rather dense and sometimes difficult to understand. Overall, results are described too much from a mathematical perspective, in terms of the performance of the logistic model, and less from a biological perspective, i.e. these are the biological implications of the results. The authors do that from time to time, but there are still many dense and jargon-filled sentences.

As an example, in lines 281-285: “The normal GK1_sf PDF primarily coincides with a region of the MC curve where P(EB) < ~0.3. In contrast, the HF GK1_sf PDF primarily coincides with a region of the MC curve where P(EB) > ~0.4.”

In general, the authors should revise to minimize sentences like this that are all about model, and focus more on the ones that explain the significance of changes in parameters such as GK1.

(3) A related comment concerns the concept of entropy of distributions, which is mentioned a few times. The first mention of this is on line 201, in describing a specific result, without any introduction. Most readers will be left wondering: why is entropy important?

Minor comments:

(1) A couple of references are duplicates, e.g. 3 and 12, and 20 and 29

(2) In the Discussion, line 368, where the authors state that this is the first simplified model of arrhythmia dynamics, that’s not quite true. Morrotti & Grandi built a logistic model of EADs (PMID: 28116246) and Varshneya et al recently applied this approach to a study of arrhythmias caused by drugs for Covid-19 (PMID: 33205613). There is still considerable value in the present work because this is stochastic and spatial models, whereas those other studies were deterministic models of a different arrhythmia mechanism, but still.

**Have all data underlying the figures and results presented in the manuscript been provided?**

Reviewer #1: Yes

PLOS authors have the option to publish the peer review history of their article (what does this mean?). If published, this will include your full peer review and any attached files.

Reviewer #1: No

Reviewer #2: No

**Have the authors made all data and (if applicable) computational code underlying the findings in their manuscript fully available?**

Reviewer #2: Yes
---

## [Decision Letter · Decision Letter 1]

7 Jul 2021

Dear Dr. Winslow,

Thank you very much for submitting your manuscript "Estimating Ectopic Beat Probability with Simplified Statistical Models that Account for Experimental Uncertainty" for consideration at PLOS Computational Biology.

As with all papers reviewed by the journal, your manuscript was reviewed by members of the editorial board and by several independent reviewers. In light of the reviews (below this email), we would like to invite the resubmission of a significantly-revised version that takes into account the reviewers' comments.

We cannot make any decision about publication until we have seen the revised manuscript and your response to the reviewers' comments. Your revised manuscript is also likely to be sent to reviewers for further evaluation.

Sincerely,

Daniel A Beard

Deputy Editor

PLOS Computational Biology

Daniel Beard

Deputy Editor

PLOS Computational Biology

Reviewer's Responses to Questions

**Comments to the Authors:**

Reviewer #1: The authors have addressed all of my concerns.

Reviewer #2: Although the authors have improved the manuscript during, a few aspects unfortunately remain unclear. I am not trying to be a difficult reviewer, but I am somewhat concerned that, in its present form, only modeling specialists will appreciate the work. The field contains many quantitative physiologists who value modeling but do not code models themselves (think David Eisner, Karin Sipido, Steve Houser, Bjorn Knollmann, etc). At present, I suspect that these potential readers would be confused about the physiological (or pathophysiological) conditions being simulated by the authors, and they would therefore not really appreciate the manuscript.

Part of my own confusion stems from the fact that, based on Figures 1A, 2E, and 2F, it appears that every simulation contains at least a small DAD. This would therefore imply that every simulation results in a cellular calcium wave. But with healthy myocytes in an experimental setting, waves are only seen infrequently, even after beta-adrenergic stimulation. Therefore, conditions have been created to favor the formation of waves, and it’s not really accurate to say “close to the diastolic physiological state,” as in the reviewer responses.

I don’t mean to belabor a minor point, but since experiments generally have to create conditions that will favor the production of calcium waves and DADs, I think it’s important for the manuscript to explicitly state this. This is alluded to a little bit with the description of the increase in the RyR opening rate, but in general it requires considerable work by the reader to figure this out, as he/she will need to read the first paragraph of the results, then look at a couple different sub-sections of the Methods, and then even have to download the supplementary table to fully understand. These aspects can still be improved

Some related, more specific questions that came up while trying to understand these issues.

(1) The reader has to do a lot of work to understand how the parameters of the default cell differ from those previously used in the Walker et al study (Ref 5). The first paragraph of the results mentions the 4 fold increase in the RyR opening rate, but then lines 446-449 in the Methods lists other changes following the study from Chu et al. And then the question becomes: when multiple simulations are run with the default parameters, what percentage result in a calcium wave and DAD? As mentioned, it appears from Figures 1A, 2E, and 2F that every simulation results in a DAD.

I think this type of information is important for the reader to understand that conditions have been created that favor the occurrence of calcium waves.

(2) Relatedly, Table S1 contains useful information about the ranges over which the parameters are varied. The reader shouldn’t have to go digging for this information, it seems like it would be really easy to include these 8 numbers in the Methods. Right now Lines 514-518 just say “obtained from experimental data,” which most readers will find unsatisfying.

But even after examining Table S1, we still don’t have all the information we need. Is the “default” cell mid-way between the minimum and the maximum?

**Have the authors made all data and (if applicable) computational code underlying the findings in their manuscript fully available?**

Reviewer #1: Yes

Reviewer #2: Yes

PLOS authors have the option to publish the peer review history of their article (what does this mean?). If published, this will include your full peer review and any attached files.

Reviewer #1: **Yes: **Seth H. Weinberg

Reviewer #2: No
---

## [Decision Letter · Decision Letter 2]

6 Oct 2021

Dear Dr. Winslow,

We are pleased to inform you that your manuscript 'Estimating Ectopic Beat Probability with Simplified Statistical Models that Account for Experimental Uncertainty' has been provisionally accepted for publication in PLOS Computational Biology.

Best regards,

Daniel A Beard

Deputy Editor

PLOS Computational Biology

Daniel Beard

Deputy Editor

PLOS Computational Biology

Reviewer's Responses to Questions

**Comments to the Authors:**

Reviewer #2: This paper should be accepted.

**Have the authors made all data and (if applicable) computational code underlying the findings in their manuscript fully available?**

Reviewer #2: Yes

PLOS authors have the option to publish the peer review history of their article (what does this mean?). If published, this will include your full peer review and any attached files.

Reviewer #2: No

---

## [Editor Report · Acceptance letter]

14 Oct 2021

PCOMPBIOL-D-21-00494R2 

Estimating Ectopic Beat Probability with Simplified Statistical Models that Account for Experimental Uncertainty

Dear Dr Winslow,

I am pleased to inform you that your manuscript has been formally accepted for publication in PLOS Computational Biology. Your manuscript is now with our production department and you will be notified of the publication date in due course.

With kind regards,

Katalin Szabo
